# Determination of Hourly Distribution of *Tuta absoluta* (Meyrick) (Lepidoptera: Gelechiidae) Using Sex Pheromone and Ultraviolet Light Traps in Protected Tomato Crops

**Gui-Fen Zhang** [1,*,†], **Yi-Bo Zhang** [1,†], **Lin Zhao** [1,2], **Yu-Sheng Wang** [1], **Cong Huang** [1], **Zhi-Chuang Lü** [1], **Ping Li** [3], **Wan-Cai Liu** [3], **Xiao-Qing Xian** [1], **Jing-Na Zhao** [1,4], **Ya-Hong Li** [5], **Fang-Hao Wan** [1], **Wan-Xue Liu** [1] and **Fu-Lian Wang** [2]

[1] State Key Laboratory for Biology of Plant Diseases and Insect Pests, Key Laboratory of Invasive Alien Species Control of Ministry of Agriculture and Rural Affairs, Key Laboratory of Integrated Pest Management of Crop of Ministry of Agriculture and Rural Affairs, Center for Management of Invasive Alien Species of Ministry of Agriculture and Rural Affairs, Institute of Plant Protection of Chinese Academy of Agricultural Sciences, Beijing 100193, China; zhangyibo@caas.cn (Y.-B.Z.)
[2] College of Life Science, Yangtze University, Jingzhou 434100, China
[3] The National Agro-Tech Extension and Service Center, Beijing 100025, China
[4] College of Plant Protection, Yunnan Agricultural University, Kunming 650201, China
[5] Yunnan Plant Protection and Quarantine Station, Kunming 650034, China
* Correspondence: zhangguifen@caas.cn
† These authors contributed equally to this work.

**Abstract:** *Tuta absoluta* (Meyrick), a leafminer that damages tomato leaves, terminal buds, flowers, and fruits, is a destructive tomato pest and is responsible for 80–100% of tomato yield losses globally. Different insect species have different courtship responses and phototropic flight rhythms. Improving the trapping effects of the sex pheromone and light traps is important for constructing an IPM system for *T. absoluta*. The present study explored the hourly distribution of *T. absoluta* adults caught by the sex pheromone (on the ground) and UV light (380 nm) traps in greenhouses over 24 h. The responses of males to sex pheromone (false female) lures were detected at dawn and early morning. The responses lasted for 3 h, from 05:30 (1 h before sunrise) to 08:30 (2 h after sunrise), and 95.8% of the males were caught during this period. The peak of the male responses to the sex pheromone was detected at 07:30 (from 06:30 to 07:30, 1 h after sunrise), and 80.8% of the males were caught during this period. The flight of male (proportion of 54.3%) and female (45.7%) adults toward the UV light traps occurred from 19:30 (time of sunset) to 06:30 (time of sunrise), lasted for 11 h, and exhibited a scotophase rhythm; 97.4% of the adults were caught during this period. The peak of adults flying toward the UV light traps occurred at 21:30 (from 20:30 to 21:30, 2 h after sunset). The rhythms of males' responses to the sex pheromone and of the adults' flight toward the UV lights can help to reveal the mechanisms of chemotactic and phototactic responses and may play a significant role in constructing an IPM system for this pest.

**Keywords:** chemotaxis; IPM; mass trapping; phototaxis; rhythm; tomato leafminer





## 1. Introduction

The tomato leafminer *Tuta absoluta* (Meyrick, 1917) (Lepidoptera: Gelechiidae), native to Peru, South America, is a destructive insect pest afflicting tomato crops worldwide and can cause 80–100% of tomato yield losses in a given year [1]. This pest was first reported outside of South America in eastern Spain in late 2006 [1,2], after which it spread throughout the Mediterranean region in only three years [1]. In 2017, the tomato leafminer was detected in China [3,4], which produces the highest tomato yield in the world [5]. Tomato leafminers inflict serious damage wherever they are found [6], and those in China are no exception; more than 10 provincial areas in northwestern, southwestern, and northern China have been

invaded in less than five years and have experienced tomato yield losses of 50–85% [7,8]. In these areas, insecticides have become the preferred control strategy [9,10]. The frequent use of insecticides has caused insecticide resistance and heavy disruption of existing integrated pest management (IPM) systems, especially in newly invaded areas [6,11].

Utilizing the responses of insects to pheromones (especially sex pheromones) [12–14] or lights [15–17] is a "clean or green" form of pest control that avoids or reduces the indiscriminate use of insecticides [18,19]. In China, farmers usually learn new techniques via regular education and training provided by the five-level (i.e., national level, provincial level, prefectural level, county level, and secondary vocational level) school system, as well as from the radio and television programs led by the Central Agricultural Radio and Television School (https://www.ngx.net.cn/ (accessed on 18 March 2023)). Mass trapping techniques based on sex pheromone [20] or lights [21–23] have long been considered promising components of IPM systems [22,24–27]. They can be well-applied in both covered (protected) and open fields. *Tuta absoluta* adults are phototactic [28,29] and chemotactic [12,13,29]. Light [28,29] or sex pheromone traps can be used to monitor and control *T. absoluta* [12,13,29]. Sex pheromone traps are considered the first line of defense against *T. absoluta*, both in open fields and greenhouses, and are used for monitoring and male annihilation [25,30]. Analyses based on the integrated indices of the color of planar sex pheromone traps and the height of their placement showed that blue traps placed directly on the ground trapped *T. absoluta* most effectively [31]. In fieldwork conducted in greenhouses in southwestern China, the number of trapped male adults was 3.53-, 26.79-, 36.11-, and 44.89-fold greater when the traps were placed directly on the ground (i.e., the height of the trap was 0 cm) than when they were placed at heights of 0–20 cm, 60–80 cm, 120–140 cm, and 180–200 cm, respectively [31]. Light traps are used ever more widely, especially in regions with scarce water resources [21,27]. Compared with traps using other wavelengths of blue-violet light, a UV light (380 nm wavelength) trap had the best trapping ability and insecticidal efficacy for *T. absoluta* when lights of all wavelengths were turned on at 19:00 and off at 07:00 [26]. *Tuta absoluta* adults are mostly active at dusk and mate from 07:00 to 11:00 [1,32]; however, they are often seen flying during the day and occasionally mating in the afternoon [32].

*Tuta absoluta* has recently invaded northwestern and southwestern China. To fully harness the potential of a sex-pheromone and light-based trapping system for control, three questions must be answered: (1) When do male adults respond to the sex pheromone, i.e., false female lures? (2) Do the male and female adults fly toward the lights throughout the night with the same rhythm? (3) Are the chemotactic (i.e., response to the sex pheromone) and phototactic rhythms of male and female adults consistent? To answer these questions, we evaluated the hourly distribution of *T. absoluta* caught using the sex pheromone and light traps in the protected tomato crops in southwestern China.

## 2. Materials and Methods

### 2.1. Study Site

The study site was located at a ~15.33 ha experimental base (GPS coordinates: 24°20′28.68″ N latitude, 102°34′43.08″ E longitude, and 1677–1738 m ASL) on a hillside in Yuxi, Yunnan, in southwestern China. The experimental base produced organic fruits and vegetables that were mainly used for family picking. The fruits and vegetables were planted in greenhouses or open fields. The greenhouses faced south and were 6 m in height. The sizes of the greenhouses depended on the terrain and mostly ranged from 0.04 ha to 0.08 ha; four are 0.16 ha. Tomatoes were the main crops cultivated at the base. The hourly distribution tests were conducted in 2020, from early May to early August.

### 2.2. Tomatoes and Their Planting

Two tomato varieties were planted: the first was cherry tomatoes (with fruits of red or yellow color, mainly used as fresh fruits), and the second was a large, fresh market tomato (with large fruits of red color, mainly used as a cooking vegetable). The tomatoes were

grown in greenhouses using two methods: One was potted planting, i.e., the tomato plants were grown in round pots (outer, inner, and bottom measurements of, respectively, 48.0, 41.5, and 26.5 cm in diameter, and 31.0 cm in height) in a tray on the ground over brickwork. The second was soil planting, i.e., the tomato plants were grown in soil covered with silver-black bicolor mulching film. The plant spacing was ~50 cm, and the row spacing was either ~120 cm (potted planting) or ~100 cm (soil planting). Tomato seedlings were cultivated at the base. Tomatoes were planted twice a year in different greenhouses in batches over two-month intervals to ensure that the tomato fruits could be harvested at any time. Tomato plants were watered regularly by drip irrigation, and no chemicals, including synthetic chemical pesticides and chemical fertilizers, were applied throughout the growth period.

*2.3. Traps*

2.3.1. Sex Pheromone Trap

The planar trap was composed of a sex pheromone lure and a sticky plate [33]. Based on previous studies [31,34], we used a sticky board (20 cm × 25 cm) in a blue color (465 ± 10 nm wavelength) (Beijing Zoje Sifang Biotechnology Co., Ltd., Beijing, China) as a capture device. The lure contained the major sex pheromone component ((3E,8Z,11Z)-3,8,11-tetradecatrien-1-yl acetate) of the *T. absoluta* female [19] and was located in the middle of the sticky plate. The synthetic (3E,8Z,11Z)-3,8,11-tetradecatrien-1-yl acetate was impregnated into rubber septa dispensers at a dosage of 500 µg/septum (PH-937-1RR, Russell IPM, Deeside, UK). The traps were placed directly on the ground (0 cm) [31] when testing the hourly distribution of *T. absoluta* caught using the sex pheromone.

2.3.2. Light Trap

In this study, slightly modified light traps were used. The ultraviolet light (UV light) traps were composed of three parts, i.e., a lamp (12 w LED in the 380 nm wavelength) [26], a waterproof sun cover (made of engineering plastics), and an insect collection water tray (34.0 cm in diameter and 4.7 cm in height) containing 0.2% detergent water to reduce the water surface tension and prevent adults from escaping from the trap, as well as to keep the trapped adults' wings spread out on the water surface. The water surface was ~1 cm from the opening of the tray. The insect-collecting water trays were placed directly on the ground [31], and the distance between the lamp end and the water surface was ~15 cm.

*2.4. Hourly Distribution Study*

2.4.1. Sex Pheromone Trap Evaluation

Five tomato greenhouses of 0.04–0.05 ha each were used for the study. The status of the tomato plants in the five greenhouses was the same, i.e., they were all ~1.8 m in height and at the beginning of fruit harvest. The pheromone traps were positioned inside the greenhouses between rows of tomato plants directly on the ground [31] near the center of the greenhouse. The hourly distribution of the number of *T. absoluta* caught by the sex pheromone traps was evaluated between early May and the end of June (sunrise, 06:28:18~06:24:52; sunset, 19:44:03~20:02:17) (https://richurimo.bmcx.com (accessed on 18 March 2023)). Since *T. absoluta* adults are active around sunrise (Zhang GF et al., personal observation) (i.e., 06:28:18~06:24:52, from here on written as 06:30), we began the field observation at the half-hour timepoint to obtain observations from around sunrise; we positioned the sex pheromone traps at 14:30, and the first counting was conducted at 15:30. The blue sticky boards were replaced once every hour. The number of trapped male adults was counted hourly. The final count was conducted the following day at 14:30. In total, 24 sets of data were obtained. The set of 24 counts was repeated five times, with about one repetition per week.

2.4.2. UV Light Trap Evaluation

Five UV light traps were positioned in the center aisles directly on the ground [31] in five tomato greenhouses. The hourly distribution of the number of *T. absoluta* adults caught

by the UV light traps was measured between the end of June and early August (sunrise, 06:24:52~06:42:39; sunset, 20:02:07~19:47:37) (https://richurimo.bmcx.com (accessed on 18 March 2023)). Since *T. absoluta* adults exhibit phototaxis [29], and sunset and sunrise occurred at approximately 19:30 and 06:30, respectively, during the experimental period, we started the field test at the half-hour timepoint to obtain observations from around sunset and sunrise, i.e., we turned on the lights at 14:30. The trapped female and male adults were then counted and removed hourly, and detergent water was added to the insect-collecting water tray to maintain an appropriate water level. The light trap test was repeated five times, with about one repetition per week. Since the numbers of other captured insect species, including moths, flies, and beetles, were all fewer than two individuals, they were not counted.

*2.5. Data Collection and Analysis*

In the sex pheromone and light-trapping greenhouses, the trapped *T. absoluta* moth adults were counted hourly, and those captured by the UV light traps were sexed based on the size, color, shape of the abdomen (female abdomen is grayish yellow in color and conical; the male abdomen is gray in color and cylindrical), as well as genitalia [3,29]. Data obtained from the hourly count of adult *T. absoluta* individuals and the proportions (in % = (number of trapped moths for each hour/total number of trapped moths for the whole 24 h) × 100) of trapped adults for each hour were analyzed with a one-way analysis of variance (ANOVA). The data were arcsine square root transformed before analysis. An LSD test was used to identify any significant differences between different hours. A paired-sample *t*-test was used to identify any significant differences between male and female adults. Means were compared at the $p < 0.05$ significance level for all parameters [35].

**3. Results**

*3.1. Hourly Distribution Evaluated for Sex Pheromone Traps*

The hourly evaluation of the average numbers and proportions indicated that the males responded to the sex pheromone, i.e., false female lures, in a very concentrated period of time, which lasted for only 3 h. During this period, the hourly numbers and proportions were the highest of the whole 24 h of trapping ($p < 0.001$). The peak of the males' responses to the sex pheromone lure was detected at 07:30 (from 06:30 to 07:30, 1 h after sunrise) (Figure 1). The hourly average numbers and proportions of male *T. absoluta* adults evaluated for the sex pheromone traps are shown in Figure 1.

3.1.1. Hourly Distribution of Male Numbers

The period of time during which the male *T. absoluta* adults responded to the sex pheromone (false female) lures was concentrated and only lasted for 1 h, i.e., from 06:30 (time of sunrise) to 07:30 (1 h after sunrise), with a peak counting value at 07:30 ($F_{23,96} = 5.585$, $p < 0.001$; one-way ANOVA, LSD test) (Figure 1A). The number of caught male adults (average of 76.2 ± 31.6 male individuals per trap) was the highest ($p < 0.001$) at 07:30 across the 24 h of hourly counting. Although 6.6 ± 3.8 and 10.4 ± 5.6 male individuals were caught at 06:30 (i.e., the trapping time from 05:30 to 06:30, 1 h before sunrise) and 08:30 (from 07:30 to 08:30, 2 h after sunrise), respectively, which were more than those caught during other hourly periods (0.0~1.0 individuals, except for 07:30), no significant difference was detected between them ($p > 0.05$) (Figure 1A). More than five male individuals were trapped during three hourly periods, i.e., 05:30–06:30 (1 h before sunrise), 06:30–07:30 (1 h after sunrise), and 07:30–08:30 (2 h after sunrise). Both before and after these three periods of time, few male individuals (0.0~1.0 individuals) were caught (Figure 1A).

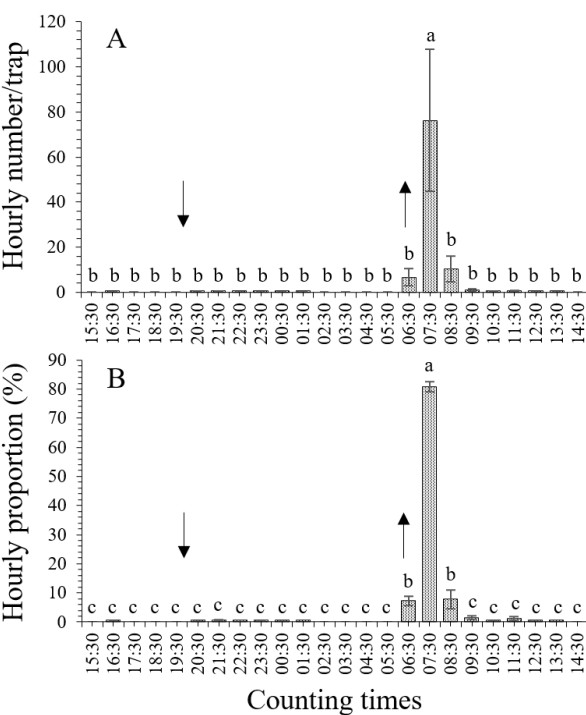

**Figure 1.** Hourly distribution of *Tuta absoluta* male adults responding to sex pheromone lures. Different lowercase letters indicate significant differences among different hourly periods at the $p < 0.05$ significance level (one-way ANOVA and LSD test). The up and down arrows indicate sunrise and sunset, respectively. (**A**), hourly distribution of numbers; (**B**), hourly distribution of proportions. The same applies in Figure 2 below.

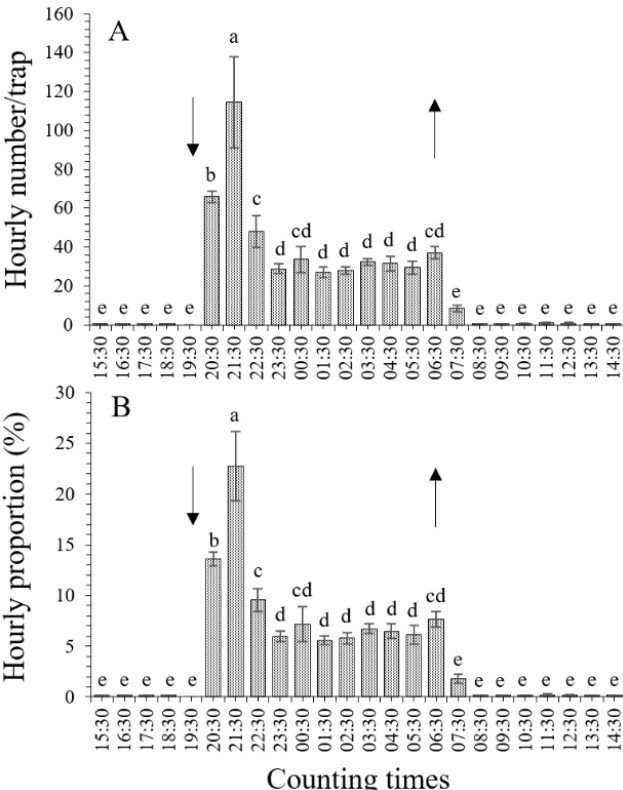

**Figure 2.** Hourly distribution of total *Tuta absoluta* adults flying toward UV light traps. Different lowercase letters indicate significant differences among different hourly periods at the $p < 0.05$ significance

level (one-way ANOVA and LSD test). The up and down arrows indicate sunrise and sunset, respectively. (**A**), hourly distribution of numbers; (**B**), hourly distribution of proportions.

### 3.1.2. Hourly Distribution of Proportions

When the ratios of the number of males caught in each hourly period to the total number of males caught during the whole 24 h period were calculated and compared, significant differences were detected ($F_{23,96} = 388.172$, $p < 0.001$) (Figure 1B). For all the trapped male adult individuals, the highest proportion, i.e., 80.8 ± 1.7%, was detected at 07:30 (i.e., 06:30–07:30, 1 h after sunrise) ($p < 0.001$). The hourly proportions at 06:30 (i.e., 05:30–06:30, 1 h before sunrise) and 08:30 (i.e., 07:30–08:30, 2 h after sunrise) were 7.2 ± 1.6% and 7.8 ± 3.3%, respectively, which were significantly higher ($p < 0.001$) than those during the other hourly periods (0.0%~1.4%), except for 07:30. During these three periods, i.e., 05:30–06:30, 06:30–07:30, and 07:30–08:30, about 95.8% of the male adults were caught. Before and after these three periods of time, the hourly proportions decreased significantly ($p < 0.001$), with hourly proportions being lower than 1.5% (Figure 1B).

### 3.2. *Hourly Distribution Evaluated for UV Light Traps*

An analysis of the hourly distribution, represented by the average numbers and proportions of *T. absoluta* adults (including total, male, and female adults), indicated that the leafminer moths flew toward the UV light traps throughout the night, i.e., from sunset (19:30) to sunrise (06:30). However, phototaxis varied significantly with differences in hourly periods ($p < 0.001$). The peak of adult flight toward the UV light traps was detected at 21:30 (from 20:30 to 21:30, 2 h after sunset). Fewer than 1.0 adult individuals (except for 8.4 ± 1.8 moths from 06:30 to 07:30, 1 h after sunrise) were caught during every hourly period of the day (from 08:30 to 19:30). The hourly average numbers and proportions of *T. absoluta* adults evaluated for the UV light traps are shown in Figures 2 and 3.

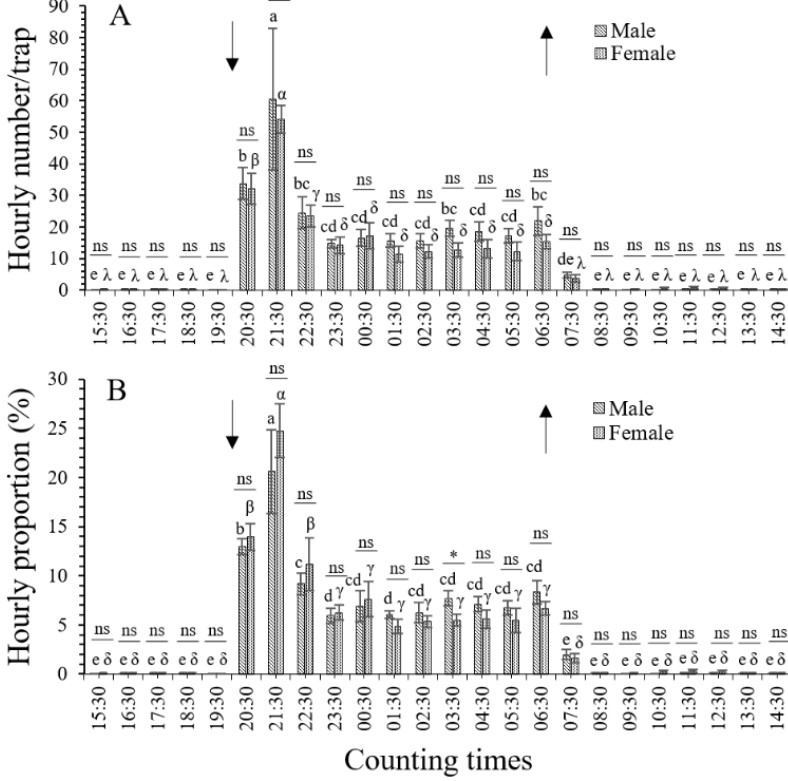

**Figure 3.** Hourly distributions of male and female *Tuta absoluta* adults flying toward UV light traps. Different lowercase Latin and Greek letters indicate significant differences among hourly periods for

male and female adults, respectively, at the $p < 0.05$ significance level (one-way ANOVA and LSD test). The up and down arrows indicate sunrise and sunset, respectively. (**A**), hourly distribution of numbers; (**B**), hourly distribution of proportions. The asterisk above the horizontal line indicates a significant difference between males and females during the same period at the $p < 0.05$ significance level (paired-sample *t*-test); ns, not significant.

### 3.2.1. Hourly Distribution of Total Adults

*Tuta absoluta* adults flew toward the UV light traps throughout the night, i.e., from sunset (19:30) to sunrise (06:30). The total numbers of *T. absoluta* adults (including male and female adults) caught by the UV light traps significantly differed among different hourly periods ($F_{23,96} = 25.036$, $p < 0.001$) (Figure 2A). The total number of adults caught on average ($114.6 \pm 23.5$ individuals per light trap) was the highest from 20:30 to 21:30 (2 h after sunset), followed by the periods from 19:30 to 20:30 (1 h after sunset) ($65.9 \pm 3.0$ adults), 21:30 to 22:30 (3 h after sunset) ($47.9 \pm 8.2$ adults), 22:30 to 23:30 (with $28.9 \pm 2.6$ adults), and every hourly period from 01:30 to 05:30 (with 27.1–32.4 adults). The lowest numbers were observed in the hourly periods from 07:30 to 19:30 (0.0–8.4 adults). Moreover, no significant differences were detected between the periods of 22:30 ($47.9 \pm 8.2$ adults), 00:30 ($33.6 \pm 6.6$ adults), and 06:30 ($37.2 \pm 3.3$ adults); among every hourly period from 23:30 to 06:30 (27.1–37.2 adults); or in any hourly period from 07:30 (1 h after sunrise) to 19:30 (time of sunset) (0.0–8.4 adults) (Figure 2A). Although a few ($8.4 \pm 1.8$) adults were attracted by the UV light traps from 06:30 to 07:30 (1 h after sunrise), this was significantly lower than during any other hourly period of the night, from 19:30 (sunset) to 06:30 (sunrise) (Figure 2A). The hourly proportions of adults, estimated by calculating the ratio of *T. absoluta* adults caught every hour to the total number of adults over the whole 24 h, were significantly different among different hourly periods ($F_{23,96} = 37.945$, $p < 0.001$) (Figure 2B). Furthermore, the distribution trends of the hourly proportions and adult numbers were completely consistent (Figure 2A,B).

### 3.2.2. Hourly Distribution of Male Adults

The number of *T. absoluta* male adults caught by the UV light traps was significantly different among different hourly periods ($F_{23,96} = 8.423$, $p < 0.001$) (Figure 3A). The number of caught male adults (average of $60.5 \pm 22.5$ individuals per light trap) was the highest from 20:30 to 21:30 (2 h after sunset), followed by from 19:30 to 20:30 (1 h after sunset) ($33.8 \pm 5.1$ males), every hourly period from 23:30 to 02:30 (14.7–16.4 males), and from 04:30 to 05:30 (17.1–18.5 males). The lowest numbers of adult males were observed in the hourly periods from 07:30 to 19:30 (0.0–4.8 males). No significant differences were detected among 20:30 ($33.8 \pm 5.1$ males), 22:30 ($24.4 \pm 11.4$ males), 03:30 ($19.7 \pm 2.6$ males), and 06:30 ($21.9 \pm 4.5$ males); in any hourly period from 22:30 to 06:30 (14.7–24.4 males); or in any hourly period from 07:30 (1 h after sunrise) to 19:30 (time of sunset) (0.0–4.8 males) (Figure 3A). Male *T. absoluta* adults flew toward the UV light traps throughout the night, with numbers peaking at 21:30 (from 20:30 to 21:30, 2 h after sunset) (Figure 3A). The hourly proportions of male adults were significantly different among different hourly periods ($F_{23,96} = 23.969$, $p < 0.001$) (Figure 3B). Moreover, the distribution trends of hourly proportions and male adult numbers were similar but had a few differences (Figure 3A,B).

### 3.2.3. Hourly Distribution of Female Adults

The number of *T. absoluta* female adults caught by the UV light traps was significantly different among different hourly periods ($F_{23,96} = 34.302$, $p < 0.001$) (Figure 3A). The number of caught female adults (average of $54.1 \pm 4.3$ individuals per light trap) was the highest from 20:30 to 21:30 (2 h after sunset), followed by that from 19:30 to 20:30 (1 h after sunset) ($32.1 \pm 4.9$ females), then from 21:30 to 22:30 (3 h after sunset) ($23.5 \pm 3.4$ females), then every hourly period from 23:30 to 06:30 (11.4–17.2 females), and every hourly period from 07:30 (1 h after sunrise) to 19:30 (time of sunset). During daylight, the UV light traps had the lowest female adult numbers (0.0–3.7 females) (Figure 3A). The hourly proportions of

female adults were significantly different among different hourly periods ($F_{23,96}$ = 33.654, $p < 0.001$) (Figure 3B). In addition, the distribution trends of hourly proportions and female adult numbers were similar but had a few differences (Figure 3A,B).

### 3.2.4. Comprehensive Analysis

The number of male adults flying toward the UV light traps exceeded that of female adults during the whole night period ($t$ = 4.454, df = 11, $p$ = 0.001; paired-sample $t$-test). No significant differences were detected between the numbers of male and female adults for any hourly period ($p > 0.05$) (Figure 3A). Furthermore, the proportions of female adults were 1.06–1.22-fold higher than those of male adults during the first half of the night, i.e., from 1 h after sunset (20:30) to midnight (00:30), although the difference was only marginally significant ($t$ = 2.404, df = 4, $p$ = 0.074; paired-sample $t$-test) (Figure 3B). However, during the second half of the night, from 01:30 to 06:30 (time of sunrise), higher proportions were detected in male adults (1.17–1.41-fold higher) than in female adults ($t$ = 7.822, df = 5, $p$ = 0.001). Additionally, the proportion of male adults was significantly higher than that of female adults at 03:30 ($t$ = 3.143, df = 4, $p$ = 0.035; paired-sample $t$-test) (Figure 3B).

## 4. Discussion

Currently, the so-called "green" prevention and control techniques are a major component of agricultural pest management systems [36]. Mass trapping techniques, especially those based on light, color, or pheromones, are important elements of "green" prevention and control techniques for agricultural insect pests [22,28,37,38]. Tomatoes are highly valued for their quality of being eaten raw, cooked, or processed but are vulnerable to attack from pests. Green techniques for insect pest control are recommended for organic or green tomato production [39,40].

In our field study, conducted using sex pheromone traps (blue in color and placed directly on the ground, i.e., the distance between the trap and the ground was 0 m), *T. absoluta* male adults responded to the sex pheromone (false female) lures from 1 h before sunrise (05:30) to 2 h after sunrise (08:30), i.e., they displayed photophasic and crepuscular behaviors for a total of 3 h. About 95.8% of the total trapped males were caught during this 3 h period, showing that this time period was the most active time for males to respond to female cues. This confirms our previous field observations (mating flight behavior occurring around sunrise in greenhouses) and the morning mating behaviors observed under greenhouse and laboratory conditions by Uchôa-Fernandes et al. [32] and Lee et al. [41], respectively. However, during other hourly periods, one or fewer male adults (with a proportion of less than 1.5%) observed responding to the sex pheromone lures could be random rather than intentional. *Tuta absoluta* male adults typically responded to female cues at dawn and in the early morning (peaking at 1 h after sunrise); the courtship behavior of female adults likely occurs synchronously, while mating behaviors should begin after or at the same time [32,41,42].

When the hourly distribution was evaluated for UV light (380 nm) traps (placed directly on the ground, i.e., the distance between the lamp and water surface and the ground was ~15/20 cm), the results indicated that the *T. absoluta* adults flew toward the UV light throughout the night, which started at sunset (19:30) and ended at sunrise (06:30), i.e., they displayed scotophasic and crepuscular behaviors. About 97.4% of the total trapped adults were caught during this period, with a peak at 2 h after sunset. Moreover, the number of male adults caught by the light trap exceeded that of female adults during the whole night, demonstrating that the phototactic response of males was stronger than that of females during our test period from the end of June to early August. A higher proportion of male than female adults ($p$ = 0.001) was detected in the second half of the night (from 01:30 to 06:30), which could be related to the following (i.e., 06:30–07:30) high proportion response to the sex pheromone traps. The lower proportion of female adults ($p$ = 0.001) observed in the same period of time might be related to sex pheromone production or biosynthesis in the so-called female "calling" behavior [42].

In a greenhouse study conducted by Uchôa-Fernandes et al., female "calling" and mating behaviors were detected during the first 2 h of photophase (from 05:45 to 08:20) [32]. This was also demonstrated by our present research, where males responded to the sex pheromone (false female) lures from 05:30 to 08:30. Female *T. absoluta* adults oviposit more frequently during the photophase (12:00–15:00, 15:00–18:00, and 18:00–21:00) based on results from a study conducted from 06:00 to the following 06:00 (observed once every 3 h), and with a peak proportion of 56.4% from 15:00–18:00 [32]. These results showed that mating happened toward the sunrise, and oviposition occurred during the day, i.e., diurnal and crepuscular behaviors were observed. The frequent flights of *T. absoluta* adults detected after sunset might have been caused by the UV light (380 nm) prolonging the nocturnal foraging activity of *T. absoluta* adults [43]. Moreover, physiological stress responses induced by light stress and interference could have also played an important role [38,44]. However, for developing more effective, ecofriendly, and economical light traps, the mechanisms of phototactic behavior should be further studied.

Courtship, mating, and oviposition are the most important life activities of adult sexually reproducing insects. Related behaviors, especially the response to sex pheromone lures (or female cues) [42] and flying toward light traps [22,38], vary between species, even between those in the same family, e.g., the potato tuberworm moth, *Phthorimaea operculella* (Lepidoptera: Gelechiidae) [45]. Males responding to sex pheromone (false female) lures and male and female adults flying toward light traps displayed obvious diel rhythms. The period of time during which the male adults responded to the sex pheromone lures was very concentrated, having occurred from 1 h before sunrise to the first 2 h after sunrise, as confirmed in our present research. This finding is significant for developing techniques based on sex pheromones, including the timed release of sex pheromones for the mass trapping of *T. absoluta* or disrupting its mating [46]. This can help save money and improve the efficacy of the prediction, monitoring, and controlling of this pest. Although polyandry or multimating has direct benefits for increasing the fecundity, fertility, and longevity of *T. absoluta* females [27,41], mass trapping effects based on sex pheromones must be improved, assuming that sex pheromone traps are properly used, especially regarding the height of traps [31].

Both male and female adults can fly toward a light source [47]. The tomato leafminer *T. absoluta* is a nocturnal insect. In our study, adults were caught by the UV light traps throughout the night (from 19:30 to 06:30), 45.7% of which were females. Of the total number of adults caught, about 97.4% were caught during this 11 h period. If the lights were turned off 1 h later (i.e., the lights were on from 19:30 to 07:30), the proportion of caught adults increased to 99.2%. Clarifying the phototactic rhythms of *T. absoluta* can benefit the research and development of new UV light traps that can effectively control this notorious pest, such as precision photosensitive UV light traps. Mass trapping based on a UV light trap should be effective, with both male and female adults caught at the same time. Although a higher proportion of males was detected in this study, higher proportions of females [26,29], including gravid females [26], had also been caught since the sex ratio of the caught adults varied over time [48].

Clarifying the chemotactic and phototactic rhythms of *T. absoluta* in response to the sex pheromone and UV light traps is important for studying the relationships between chemotaxis, phototaxis, and circadian rhythms. Furthermore, this can provide insight into the mechanisms by which mating and phototactic behaviors are regulated and can benefit pest management [42].

## 5. Conclusion

This study demonstrated that the chemotactic (i.e., response to sex pheromone lures) and phototactic rhythms of male and female adults were not consistent. The male adults responding to the sex pheromone (false female) lures for mating exhibited an early morning rhythm, with responses that lasted for 3 h and peaked at 07:30 (from 06:30 to 07:30, 1 h after sunrise). The responses of male adults at particular times could help with the timed

release of sex pheromones for mass trapping and mating disruption in order to save costly pheromones and increase longevity. Male and female adults exhibited scotophasic rhythms and flew toward the UV light traps throughout the night (from 19:30 to 06:30). Improved mass trapping based on sex pheromones and UV lights has the potential to control *T. absoluta* in greenhouses and can be integrated with other components of IPM programs to effectively manage this pest.

**Author Contributions:** Conceptualization, G.-F.Z. and Y.-B.Z.; methodology, G.-F.Z., Y.-B.Z., L.Z., P.L., W.-C.L., J.-N.Z. and Y.-H.L.; formal analysis, G.-F.Z., Y.-B.Z., Y.-S.W., C.H. and L.Z.; data calculation, G.-F.Z., Y.-B.Z., L.Z., Z.-C.L. and C.H.; statistical analysis, G.-F.Z., Y.-B.Z., L.Z. and X.-Q.X.; writing—original draft preparation, G.-F.Z., Y.-B.Z. and L.Z.; writing—review and editing, P.L., W.-C.L., W.-X.L., F.-L.W. and F.-H.W.; visualization, G.-F.Z., Y.-B.Z. and L.Z.; supervision, G.-F.Z. and Y.-B.Z. G.-F.Z. and Y.-B.Z. contributed equally to this work. All authors have read and agreed to the published version of the manuscript.

**Funding:** This work was supported by the National Key Research and Development Project of China (grant numbers 2021YFD1400200, 2017YFC1200600, and 2021YFC2600400) and the Science and Technology Innovation Program of the Chinese Academy of Agricultural Sciences (grant number caascx-2019-2023-IAS, caas-zdrw202203).

**Institutional Review Board Statement:** Not applicable.

**Informed Consent Statement:** All authors have given consent to publish this article.

**Data Availability Statement:** All data are included and available in the article.

**Acknowledgments:** The authors are grateful to the Ministry of Science and Technology of the People's Republic of China and the Chinese Academy of Agricultural Sciences for providing financial support; the State Key Laboratory for Biology of Plant Diseases and Insect Pests, the Key Laboratory of Invasive Alien Species Control, and the Ministry of Agriculture and Rural Affairs; and the Yuxi Yangrui Organic Vegetable Production Base of the Institute of Plant Protection of the Chinese Academy of Agricultural Sciences for the research facilities.

**Conflicts of Interest:** The authors declare no conflict of interest.

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
