# Peer review of "Determination of Hourly Distribution of Tuta absoluta (Meyrick) (Lepidoptera: Gelechiidae) Using Sex Pheromone and Ultraviolet Light Traps in Protected Tomato Crops"

_horticulturae, doi:10.3390/horticulturae9030402_

Round 1

Reviewer 1 Report

Suggestions are attached with the MS.

Reviewer 2 Report

This study evaluated tomato leafminer pheromone and UV traps efficacy based on time throughout the day. They did this by evaluating trap catches every hour for 24 h in 5 tomato greenhouses in China.

Major comments:

The introduction was lacking a lot of context that was then mentioned in the discussion section. Sections of the discussion such as current knowledge of tomato leafminer behavior and optimal trap placement should be put in the introduction to give background on IPM development and why particular methods were used in this study.

The results section was very redundant. The count data and the proportion data showed essentially the same conclusions, and could be cut in half. Though I think keeping the male and female data for the UV light traps separate was useful.

I would also like to see more discussion of why flights were so common after sunset if mating is happening towards sunset and oviposition is occurring during the day.

Line by line comments:

There were no line numbers, so precise comments are hard to give. So, I will try to list them by section/page number.

Page 1, highlight 1: change to “…build the Tuta absoluta IPM system.”

The abstract was missing some context in the project setup area. Where is this pest a problem? Globally? Add some brief information here about what kind of damage the pest does.

In the first paragraph of the introduction: Provide more detail on the importance of tomato as a crop if the first sentence is going to be included.

In the first paragraph of the introduction: Add the order and family information for the first mention of the pest again here.

In the first paragraph of the introduction: Is South America the native range?

Second sentence of the second paragraph of the introduction: can remove “measure” from the sentence. Would also like to see a greater explanation of how trapping contributes to IPM plans for those not proficient in their use.

Second paragraph of the introduction: Would like to see a lot more introduction to what is known about the pest and the current trapping methods here.

Third line of the third paragraph of the introduction: change “…1) when does the male…”

Lines 4-5 of the third paragraph of the introduction: Why do you want to examine night-flying behavior here if current knowledge of the pest indicates that they are active during the day (as stated in the prior paragraph of the introduction)?

Last line on page 3: change sentence into past tense to match the rest of the section.

Methods: I was confused why you would place the trap on the ground until the discussion section. Make sure to cite that reference again in the methods.

Methods: Please, clarify the replication. Was each test done in each greenhouse for one period of 24 hours with 1 sample every hour during that time?

Line 5 in the “evaluated by UV-light traps section on page 4: change to “…adults exhibit phototaxis…”

First sentence of the discussion section: specify “green” as being methods other than chemical pesticides (I am assuming).

First paragraph of the discussion section: Remove sentence on tomatoes as a fruit and vegetable. But provide references for the next sentence on green techniques for pest control being recommended. The rest of this paragraph should be moved/incorporated into the introduction.

Note throughout: I would recommend switching the phrasing of “female calls” to maybe “cues” since “calling” often reminds people of sounds rather than chemical communication.

First sentence, third paragraph on page 10: remove “bisexual” from the first sentence.

Third paragraph on page 10: the middle part of this paragraph was redundant with the previous paragraph.

Reviewer 3 Report

Reviewer

Comments to Author

The manuscript title is “he hourly distribution of Tuta absoluta (Meyrick) (Lepidoptera: Gelechiidae) evaluated by sex pheromones and ultraviolet-light traps in protected tomato crops” by Gui-Fen Zhang et al. The hourly distribution of T. absoluta caught by sex pheromone and light traps in protected tomato crops in southwestern China was evaluated in this study. This manuscript doesn't meet the standards of science, and it has many technical, basic, grammatical, and typographical errors. As a result, I cannot recommend it for publication.

Major points

  1. The manuscript title is very hard to understand, so please revise it.
  2. Please remove the highlight points from this journal; they are unnecessary.
  3. Please use line numbers; it will be helpful for reviewing the manuscript.
  4. The introduction part is poorly written and currently not up to the scientific standard; it also looks like a textbook, so please carefully revise it.
  5. The research objectives are unclear at the end of the introduction.
  6. In the results part, the data interpretation is very poor and hard to understand, so please carefully revise it.
  7. In the discussion part, please don’t use your results, this is not the results part.
  8. The conclusion part is not sufficient; please write elaborately.

Round 2

Reviewer 3 Report

Comments to the authors

The authors corrected the errors that reviewers had mentioned in the manuscript. The current form of the manuscript is worthy of publication.